# Volumetric Correspondence Networks for Optical Flow

**Gengshan Yang**[1][*], **Deva Ramanan**[1,2]
[1]Carnegie Mellon University, [2]Argo AI
{gengshay, deva}@cs.cmu.edu

## Abstract

Many classic tasks in vision – such as the estimation of optical flow or stereo disparities – can be cast as dense correspondence matching. Well-known techniques for doing so make use of a *cost volume*, typically a 4D tensor of match costs between all pixels in a 2D image and their potential matches in a 2D search window. State-of-the-art (SOTA) deep networks for flow/stereo make use of such volumetric representations as internal layers. However, such layers require significant amounts of memory and compute, making them cumbersome to use in practice. As a result, SOTA networks also employ various heuristics designed to limit volumetric processing, leading to limited accuracy and overfitting. Instead, we introduce several simple modifications that dramatically simplify the use of volumetric layers - (1) volumetric encoder-decoder architectures that efficiently capture large receptive fields, (2) multi-channel cost volumes that capture multi-dimensional notions of pixel similarities, and finally, (3) separable volumetric filtering that significantly reduces computation and parameters while preserving accuracy. Our innovations dramatically improve accuracy over SOTA on standard benchmarks while being significantly easier to work with - training converges in 7X fewer iterations, and most importantly, our networks generalize across correspondence tasks. On-the-fly adaptation of search windows allows us to repurpose optical flow networks for stereo (and vice versa), and can also be used to implement adaptive networks that increase search window sizes *on-demand*.

## 1 Introduction

Many classic tasks in vision – such as the estimation of optical flow [13] or stereo disparities [34] – can be cast as dense correspondence matching. Well-known techniques for doing so make use of a *cost volume*, typically a 4D tensor of match costs between all pixels in a 2D image and their potential matches in a 2D search window. State-of-the-art (SOTA) deep networks for stereo can make use of 3D volumetric representations because the search window reduces to a epipolar line [11, 22]. Search windows for optical flow need to be two-dimensional, implying that cost volumes have to be 4D. Because of the added memory and compute demands, deep optical flow networks have rarely exploited volumetric processing until recently. Even then, most employ heuristics that reshape cost volumes into 2D data structures that are processed with 2D spatial processing [7, 18, 19, 39, 42].

Specifically, common workarounds reshape a 4D array $(x, y, u, v)$ into a multichannel 2D array $(x, y)$ with $uv$ channels. This allows for use of standard 2D convolutional processing routines, but implies that feature channels are now tied to particular $(u, v)$ displacements. This requires the network to memorize particular displacements in order to report them at test-time. In practice, such networks are quite difficult to train because they require massive amounts of data augmentation and millions of training iterations to effectively memorize [7, 19].

---

[*]Code will be available at github.com/gengshay-y/VCN.

We introduce three simple modifications that significantly improve performance and generalizability by enabling *true* volumetric processing of cost volumes:

1. We propose the 4D volumetric counterpart of 2D encoder-decoder "U-Net" architectures, which are able to efficiently encode large receptive fields for cost volume processing.

2. We propose multi-channel cost volumes that make use of multiple pixel embeddings to capture complementary notions of similarity (or match cost). We demonstrate that these multiple matches allow for better handling of ambiguous correspondences, which is particularly helpful for ambiguous coarse matches in a coarse-to-fine matching network [38].

3. We implement 4D convolutional kernels with separable high-order filters. In particular, our separable factorization results in a spatial $(x, y)$ filter that enforces spatial regularity of a flow feild, and an inhibitory "winner-take-all" or WTA $(u, v)$ filter that competes candidate matches for a given $(x, y)$ pixel.

Our innovations dramatically improve accuracy over SOTA on standard flow benchmarks while being significantly easier to work with - training converges in 7X fewer iterations. Interestingly, our networks appear to generalize across diverse correspondence tasks. On-the-fly adaptation of search windows allows us to repurpose optical flow networks for stereo (and vice versa), and can also be used to implement adapative networks that increase search window sizes *on-demand*. We demonstrate the latter to be useful for stereo matching with noisy rectifications.

## 2 Related Work

**Dense visual correspondence**    Finding dense pixel correspondences between a pair of images has been studied extensively in low-level vision. Concrete examples include stereo matching and optical flow [13, 34]. Stereo matching constrains the search space to a horizontal scanline, where a 3D cost volume is usually built and optimized to ensure global consistency[11, 23]. Though optical flow with small motion has been well-addressed by the classific variational approaches [37], finding correspondences in the 2D target image remains a challenge when displacements are large and occlusion occurs [3].

**Correspondence matching with cost volume**    Classic stereo matching algorithms usually extract local patch features and create a regular 3D cost volume, where smoothness constraints are further enforced by energy minimization [13, 34]. Recently, hand-crafted feature extraction is replaced with convolutional networks and cost-volume optimization step is commonly substituted by 3D convolutions[22, 28, 45]. Despite their similar formulation, "true" 4D cost volume is rarely used in optical flow estimation until very recently. Xu et al. [42] directly construct and process a 4D cost volume using semi-global matching. Recent successful optical flow networks also build a correlation cost volume and process it with 2D convolutions [7, 24, 39]. There also exists work in semantic correspondence matching on a 4D cost volume with 4D convolutions [31].

**Efficient convolutional networks** Recent years have seen great interest in designing computation-efficient and memory-friendly deep convolutional networks. At operation level, depthwise separable convolutions [36] save computations by separating a multi-channel 2D convolution into a depthwise convolution and a pointwise convolution [6, 14, 33, 46]. Efforts have also been made in using tensor factorization to speed up a trained network [20, 25]. Different from prior works, we separate a 4D convolution kernel into two separate 2D kernels. At architecture level, U-Net encoder-decoder scheme is widely used in dense prediction task [1, 7, 32]. Instead of directly filtering the high-res feature maps, it saves memory and computation by downsampling the input feature maps with strided convolutions and upsampling them back. Typically, it is able to acquire sufficient receptive fields with very few numbers of layers. Similarly, we downsample the 4D cost-volume in (u,v) dimension to maintain a small memory footprint.

## 3 Approach

In this section, we first introduce a 4D convolutional matching module for volumetric correspondence processing. We then show by factorizing the filter into separable components that are implemented with an encoder-decoder [32], one can significantly reduce computation and memory. Finally, we

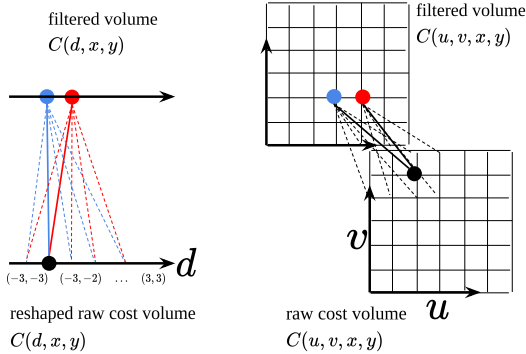

filtered volume
$C(d, x, y)$

filtered volume
$C(u, v, x, y)$

$d$

$(-3, -3)$  $(-3, -2)$  ...  $(3, 3)$

$v$

$u$

reshaped raw cost volume
$C(d, x, y)$

raw cost volume
$C(u, v, x, y)$

Figure 1: We compare 2D filtering of a 4D volume reshaped to be a multi-channel 2D array (**left**) versus true 4D filtering (**right**). For simplicity, we visualize the candidate $7 \times 7$ array of $(u, v)$ match costs for a particular $(x, y)$ pixel. Blue and red circles indicate filtered values, and lines connected to them indicate filter weights between two layers. Note that 2D filter weights are not shared across spatial locations (indicated by different colors), while 4D filter weights are. During gradient-based learning of the 2D filter, a particular observed $(u, v)$ displacement only backprops along the particular colored weights connected to it. On the other hand, the 4D filter will be updated for any observed $(u, v)$ displacement, making it easier to generalize to different displacements.

integrate volumetric filtering into a coarse-to-fine warping scheme [18, 39], where ambiguous matches and coarse-mistakes are handled by the multi-hypotheses design.

## 3.1   4D Convolutional Matching Module

Let $\mathbf{F}^1, \mathbf{F}^2 \in R^{d \times H \times W}$ be the $d$-dimensional pixelwise embedding of the source and target image. We construct a 4D cost volume by computing the cosine similarity between each pixel in the $H \times W$ source image with a set of candidate targets in a $U \times V$ search window:

$$C(\mathbf{u}, \mathbf{x}) = \frac{\mathbf{F}^1(\mathbf{x}) \cdot \mathbf{F}^2(\mathbf{x} + \mathbf{u})}{||\mathbf{F}^1(\mathbf{x})|| \cdot ||\mathbf{F}^2(\mathbf{x} + \mathbf{u})||}, \quad C(\mathbf{u}, \mathbf{x}) \in R^{U \times V \times H \times W},$$

where $\mathbf{x} = (x, y)$ is the source pixel coordinate and $\mathbf{u} = (u, v)$ is the pixel displacement. Cosine similarity is used in person re-identification and face verification [27, 41] in replacement of dot product, and empirically we find it produces a better result over dot product.

**2D convolution vs 4D convolution** Many recent optical flow networks re-organize the 4D cost volume into a multichannel 2D array with $N = U \times V$ channels, and process it with multi-channel 2D convolutions [7, 18, 19, 39]. Instead, we leave the 4D cost volume $C(\mathbf{u}, \mathbf{x})$ as-is and filter it with 4D convolutions. Much as 2D filters ensure translation invariance and generalize to images of different sizes [26], we posit that 4D filters may ensure a form of offset "invariance" and generalize to search windows of different sizes. Fig. 1 suggests that multi-channel 2D filtering requires the network to memorize particular displacements seen during training. By explicit cost volume processing, volumetric filtering of cost volumes is preferable because 1) It significantly reduces the number of parameters and computations; 2) It is capable of processing variable-sized cost volumes on demand; 3) It generalizes better to displacements that are not seen in the training.

**Truncated soft-argmin** Given a (filtered) cost volume, one natural approach to reporting the $(u, v)$ displacement for a pixel $(x, y)$ is a "winner take all" (WTA) operation that returns the argmin displacement. Alternatively, if the offset dimensions are normalized by a softmax, one could compute the *expected* offset by taking a weighted average of offsets with weights given by the probabilistic softmax (soft argmin) [22]:

$$\mathbb{E}[\mathbf{u}] = \sum_i \mathbf{u}_i p(\mathbf{u} = \mathbf{u}_i), \qquad \text{[Soft Argmin]}$$

Unfortunately, WTA is not differentiable, while the soft argmin is sensitive to changes in the size of the search window [40]. Instead, we combine both with a "truncated soft-argmin" that zeros out the softmax probabilities for displacements more than $M$ pixels away from the argmin $\mathbf{u}^*$:

$$p'(\mathbf{u} = \mathbf{u}_i) \propto \begin{cases} p(\mathbf{u} = \mathbf{u}_i), & |\mathbf{u}_i - \mathbf{u}^*| \le M \\ 0, & \text{otherwise} \end{cases} \qquad \text{[Truncated Soft Argmin]}$$

We empirically set $M = 3$ for a $7 \times 7$ search window, and use truncated soft-argmin for training and testing. Later we show that a truncated soft-argmin produces a notable improvement over soft-argmin.

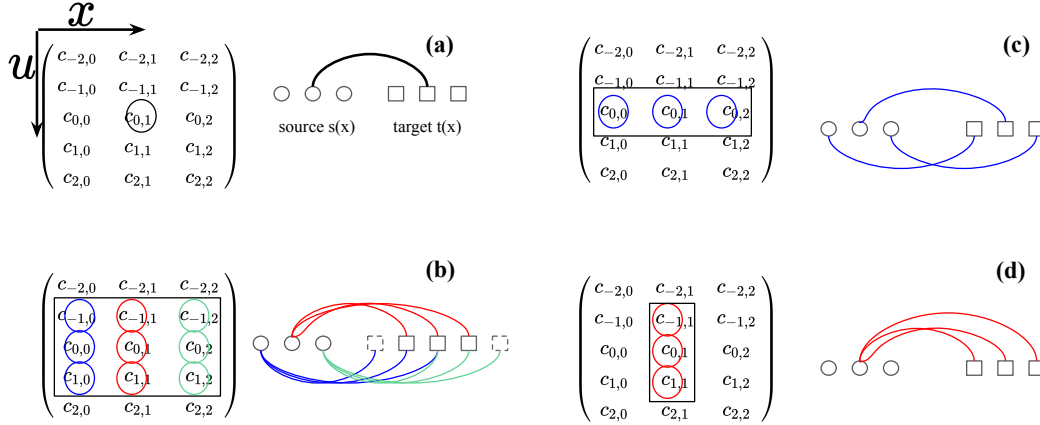

Figure 2: For ease of visualization, we show the 2D cost volume $C(u, x)$ for matching pixels across a source and target *scanline* image (**a**). To efficiently filter the volume, we factor the $3 \times 3$ filter (**b**) into a 1D spatial convolution over positions (**c**) followed by a 1D WTA convolution over displacements (**d**).

## 3.2  Efficient Cost Volume Processing

**Separable 4D convolution** We now show that 4D volumetric kernels can be dramatically simplified by factorizing into separable components. In the context of a cost volume, we propose a factorization of a 4D filter $K(\mathbf{u}, \mathbf{x})$ into a 2D spatial filter $K_S(\mathbf{x})$ and a 2D WTA $K_{WTA}(\mathbf{u})$ filter:

$$K(\mathbf{u}, \mathbf{x}) * C(\mathbf{u}, \mathbf{x}) = \sum_{\mathbf{v}, \mathbf{y}} K(\mathbf{v}, \mathbf{y}) C(\mathbf{u} - \mathbf{v}, \mathbf{x} - \mathbf{y}) \qquad \text{[4D Convolution]}$$

$$= \sum_{\mathbf{v}, \mathbf{y}} \Big[ K_{WTA}(\mathbf{v}) K_S(\mathbf{y}) \Big] C(\mathbf{u} - \mathbf{v}, \mathbf{x} - \mathbf{y}) \qquad \text{[Factorization]}$$

$$= \sum_{\mathbf{v}} K_{WTA}(\mathbf{v}) \Big[ \sum_{\mathbf{y}} K_S(\mathbf{y}) C(\mathbf{u} - \mathbf{v}, \mathbf{x} - \mathbf{y}) \Big] \qquad \text{[Separable Filtering]}$$

$$= K_{WTA}(\mathbf{u}) * \Big[ K_S(\mathbf{x}) * C(\mathbf{u}, \mathbf{x}) \Big]$$

Fig. 2 visualizes this factorization, which reduces computation by $N^2$ for a $N \times N \times N \times N$ filter with negligible effect on peformance, as shown in ablation study Tab. 4.

**U-Net encoder-decoder volume filtering** We find it important to make use of 4D kernels with large receptive feilds that can take advantage of contextual cues (as is the case for 2D image filtering). However, naively implementing large volumetric filters takes a considerabe amount of memory [22]. We found it particularly important to include context for WTA filtering. Inspired by spatial encoder-decoder networks [1, 32] we apply two downsampling layers and two upsampling layers rather than stacking multiple 4D convolutional layers. In Sec. 4.3, we show that encoder-decoder architectures allow us to significantly improve accuracy given alternatives with a similar compute budget.

## 3.3  Multi-hypotheses Correspondence Matching

**Multi-channel cost volume** Past work has suggested that cost volumes might be too restricted in size and serve as too much of an information bottleneck for subsequent layers of a network [5, 22]. One common solution in the stereo literature is the construction of a *feature* volume rather than a cost volume, where an additional dimension of feature channels is encoded in the volumetric tensor [22] - typically, one might include the difference of the two feature descriptors being compared within the cost volume, resulting in an additional channel of dimension $|\mathbf{F}(\mathbf{x})|$.

In our case, this would result in a prohibitively large volume. Instead, we propose an "intermediate" strategy between a traditional cost volume and a contemporary (deep) feature volume: a *multi-channel* cost volume. Intuitively, rather than simply encoding the cosine similarity between two embedding vectors, we record $K$ similarities between $K$ different feature embeddings that are trained jointly, by

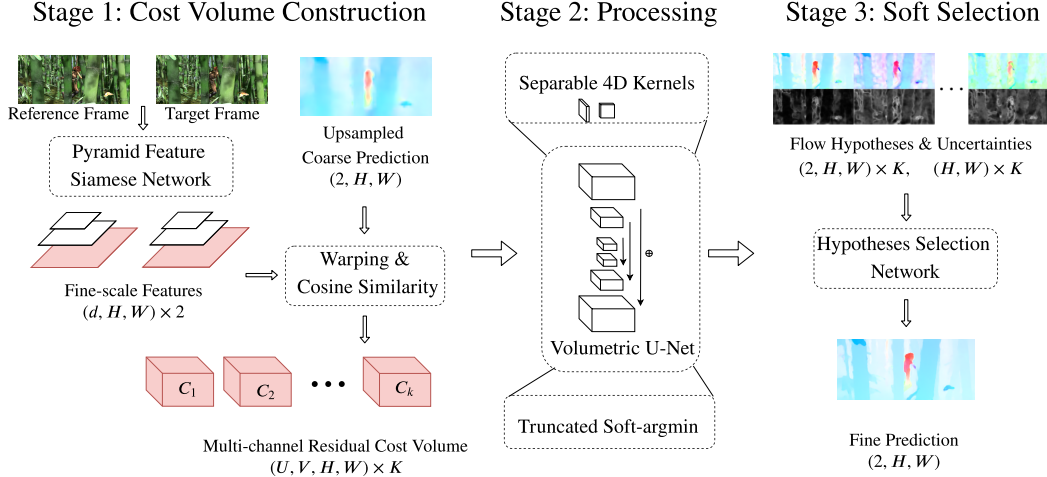

**Stage 1: Cost Volume Construction**   **Stage 2: Processing**   **Stage 3: Soft Selection**

Figure 3: Illustration of volumetric processing at one pyramid level. 1) Cost volume construction: We warp features of the target image using the upsampled coarse flow and compute a multi-channel cost volume. 2) Volume processing: The multi-channel cost volume is filtered with separable 4D convolutions, which is integrated into a volumetric U-Net architecture. We predict multiple flow hypotheses using truncated soft-argmin. 3) Soft selection: The flow hypotheses are linearly combined considering their uncertainties and the appearance feature.

taking channel-wise product between each pair of potential matches [10]. While this can be thought of as $K$ distinct cost volumes, we instead concatenate them into a multi-channel 4D cost volume $R^{K \times U \times V \times H \times W}$ where $K$ is treated as a feature channel that is kept constant in dimension during filtering. After being processed by the volumetric U-Net, each of the $K$ cost-volumes $C_k(u, v, x, y)$ is used to compute a truncated softmax expectation.

**Multi-hypotheses selection**   Considering the multimodal nature of correspondence matching, we propose a multi-hypotheses selection module that assigns weights to each hypothesis given its value, uncertainty and appearance information. Inspired by Campbell et al. [4], we treat it as a labelling problem and use a stacked 2D convolution network that takes the image features, K hypotheses values, and K entropy scores as the input, to produce a softmax distribution over the hypotheses. The final correspondence prediction is computed by weighting the hypotheses with the softmax distribution.

Coarse-to-fine warping architecture, such as PWC-Net [39], is sensitive to coarse-level failures, where the incorrect coarse flow is used to warp the features and lead to gross errors. More importantly, small objects with large displacement are never considered, since only one coarse prediction is used to warp a group of fine-pixels (usually $2 \times 2$). To account for the missing multi-modal information of the coarse scale, one solution is to create $K$ different warpings and delta fine cost volumes according to $K$ different coarse-scale hypotheses, and then aggregate the results. However, processing $K$ different hypotheses would be prohibitively expensive. Instead, we directly pass $K$ coarse-level hypothesized correspondences to the subsequent fine-scale multi-hypotheses network as additional hypotheses [43].

**Out-of-range detection**   During occlusions or severe displacements, the optimal predicted displacement is likely an "out-of-range" output that lies outside the search window. We use the processed cost volumes to train such a binary classifier. Since cost volumes allow us to access a distribution over all candidate matches, we can use the distribution to estimate uncertainty. Specifically, for each of the $K$ hypothesized cost volumes, we compute the Shannon entropy of the truncated softmax given by

$$\mathbb{H}[\mathbf{u}] = -\sum_i p'(\mathbf{u} = \mathbf{u}_i) \cdot \log p'(\mathbf{u} = \mathbf{u}_i)$$

Since Shannon entropy itself is not a reliable uncertainty indicator [15], we pass them into a U-Net module along with the image features and expected displacements, and produce a binary variable that indicates whether the ground-truth displacement is out-of search range. The out-of-range detection module is trained with binary cross-entropy loss where the supervision comes from comparing the ground-truth flow with the maximum search range. Empirically, adding the out-of-range detector regularizes the model and improves the generalization ability as shown in Sec.4.3.

Table 1: Model size and running time. Gflops is measured on KITTI-sized (0.5 megapixel) images. Number of training iterations is recorded for the pre-training stage on FlyingChairs and FlyingThings, and (S) indicates sequential training on separate modules.

| Method | #param. | Gflops | #train iter. |
|---|---|---|---|
| FlowNetS [7] | 38.7M | 66.8 | 1700K |
| FlowNetC [7] | 39.2M | 69.6 | 1700K |
| FlowNet2 [19] | 162.5M | 365.6 | 7100K (S) |
| PWC-Net+ [39] | 9.4M | 90.8 | 1700K |
| LiteFlowNet [17] | 5.4M | 151.7 | 2000K (S) |
| HD$^\wedge$3F [44] | 39.9M | 186.1 | - |
| IRR-PWC [21] | 6.4M | - | 1700K |
| Ours-small | 5.2M | 36.9 | 220K |
| Ours | 6.2M | 96.5 | 220K |

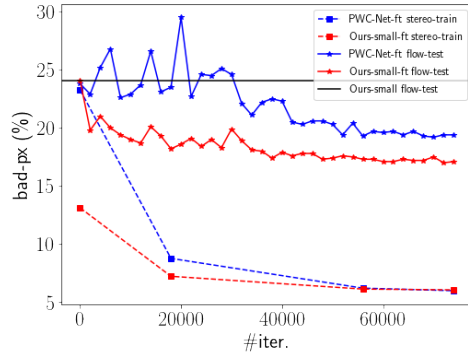

Figure 4: Stereo → Flow transfer. After fine-tuning with KITTI *stereo* data, our small model consistently out-performs PWC-Net on KITTI *flow*, though with similar error on the stereo training set, indicating our model is more generalizable.

# 4 Experiments

**Network specification** Similar to PWC-Net and LiteFlowNet [18, 39], we follow the coarse-to-fine feature warping scheme as shown in Fig. 3. We find correspondences with $9 \times 9$ search windows on a feature pyramid with stride $\{64, 32, 16, 8, 4\}$. We keep $K = \{16, 16, 16, 16, 12\}$ hypotheses at each scale. Besides the full model, we also train a smaller model that only takes features from coarse levels with stride $\{64, 32, 16, 8\}$, indicated by "Ours-small".

**Training procedure** We build the model and re-implement the training pipeline of PWC-Net+ [39] using Pytorch. The model is trained on a machine with 4 Titan X Pascal GPUs. The same training and fine-tuning procedure is followed. To be noted, we are able to stably train the network with a larger learning rate ($10^{-3}$ vs $10^{-4}$) and fewer iterations (140K vs 1200K on FlyingChairs and 80K vs 500K on FlyingThings) compared to prior optical flow networks. Furthermore, people find that PWC-Net is sensitive to initialization [39] and several attempts of training with random initialization have to be made to avoid the poor local minimum, which is never observed for our case.

## 4.1 Benchmark results

As shown in Tab. 1, our models can be trained with significantly fewer iterations without sequential training of submodules. In terms of computation efficiency, our small model only uses less than half of the FLOPS used by PWC-Net and a quarter of the FLOPS for LiteFlowNet. Our full model uses similar computation as PWC-Net and 40% fewer computations than LiteFlowNet. It is also a compact model among the ones with the least number of parameters. One more thing to notice is that our model is the only optical flow network in the table that processes a "true" 4D cost volume instead of convolving a "pseudo" multi-channel 2D cost volume.

Though our model is compact, computationally efficient and trained with fewer iterations, it demonstrates SOTA accuracy on multiple benchmarks. As shown in Tab. 2, after the pretraining stage, ours-small achieves smaller end-point-error (EPE) than all methods on KITTI [9, 30], except for LiteFlowNet2, which is heavier than LiteFlowNet, and much heavier than ours-small. Our full model further out-performs our small model and reduces the F1-all error by one-third compared to PWC-Net. On Sintel, our small model beats all previous networks except for FlowNet2, which uses 8X more computations, 30X more parameters, and 30X more training iterations. Our full model further improves the accuracy over our small model. The pretraining-stage results demonstrate that our network can generalize better than the existing optical flow architectures.

After fine-tuning on KITTI, our model clearly out-performs existing SOTA methods by a large margin. The only method comparable to ours is HD$^\wedge$3F, which uses 6X more parameters and 1.76X more computation compared to ours. On Sintel, our method ranks 1st for both the "clean" pass and the "final" pass over all two-frame optical flow methods. Noticebly, our small model achieves similar flow error on KITTI as LiteFlowNet2 and PWC-Net+ using 1/4 and 2/5 computations of theirs respectively.

Table 2: Results on K(ITTI)-15 and S(intel) optical flow benchmark. "C+T" indicates models pre-trained on Chairs and Things [7, 29]."+K/S" indicates models fine-tuned on KITTI or Sintel. $^{\dagger}$:D1-all is the default metric for KITTI stereo matching, and is evaluated on KITTI-15 stereo training data. The subscript number shows the absolute ranking among all two-frame optical flow methods in the benchmark. Best results over each group are bolded, and best results overall are underlined. Parentheses mean that the training and testing are performed on the same dataset. Some results are shown as mean $\pm$ standard deviation in five trials.

| | Method | K-15-train | | K-15-test | | S-train (epe) | | S-test (epe) | |
|---|---|---|---|---|---|---|---|---|---|
| | | Fl-epe | Fl-all | Fl-all | D1-all$^{\dagger}$ | Clean | Final | Clean | Final |
| - | FlowFields [2] | 8.33 | 24.4 | - | - | 1.86 | 3.06 | 3.75 | 5.81 |
| | DCFlow [42] | - | 15.1 | 14.83 | - | - | - | 3.54 | 5.12 |
| C+T | FlowNet2 [19] | 10.08 | 30.0 | - | - | **2.02** | **3.54** | 3.96 | 6.02 |
| | PWC-Net [38] | 10.35 | 33.7 | - | 23.30 | 2.55 | 3.93 | - | - |
| | LiteFlowNet [17] | 10.39 | 28.5 | - | - | 2.48 | 4.04 | - | - |
| | LiteFlowNet2 [18] | 8.97 | 25.9 | - | - | 2.24 | 3.78 | - | - |
| | HD$^{\wedge}$3F [44] | 13.17 | **24.0** | - | - | 3.84 | 8.77 | - | - |
| | Ours-small | $9.43 \pm 0.18$ | 33.4 | - | 13.12 | 2.45 | 3.63 | - | - |
| | Ours | **8.36** | 25.1 | - | **8.73** | 2.21 | 3.62 | - | - |
| +K/S | FlowNet2 [19] | (2.30) | (8.6) | 11.48 | - | (1.45) | (2.01) | 4.16 | 5.74 |
| | PWC-Net-ft+ [39] | (1.50) | (5.3) | 7.72 | 9.17 | (1.71) | (2.34) | 3.45 | 4.60 |
| | LiteFlowNet2-ft [18] | (1.47) | (4.8) | 7.74 | - | (1.30) | (1.62) | 3.45 | 4.90 |
| | IRR-PWC-ft [21] | (1.63) | (5.3) | 7.65$_3$ | - | (1.92) | (2.51) | 3.84 | 4.58 |
| | HD$^{\wedge}$3F-ft [44] | (1.31) | (4.1) | 6.55$_2$ | - | (1.87) | (1.17) | 4.79 | 4.67 |
| | Ours-small-ft | (1.41) | (5.5) | 7.74 | 6.10 | (1.84) | (2.44) | 3.26 | 4.73 |
| | Ours-ft | (1.16) | (4.1) | **6.30$_1$** | **4.67** | (1.66) | (2.24) | **2.81$_1$** | **4.40$_1$** |

On Sintel clean pass, our small model is better than all convolutional optical flow methods except for our full model.

Interestingly, on KITTI stereo matching training set, our method out-performs PWC-Net with an even larger margin, i.e., 8.73% error versus 23.30% without fine-tuning, and 4.67% versus 9.17% after fine-tuning on KITTI flow data. This indicates the superior generalization ability of our model across correspondence tasks.

## 4.2 Generalization ability

**Cross task generalization: Stereo $\rightarrow$ Flow** To compare the generalization ability of our method with existing deep flow networks [7, 38], we transfer the Chairs/Things-pretrained model to the real domain, i.e., KITTI, where it is difficult to acquire flow annotations than stereo (depth) annotations. To do so, we fine-tune our pretrained small model using KITTI stereo training set together with FlyingChairs and FlyingThings for 75K iterations. As comparison, a pretrained official PWC-Net model is also fine-tuned with the same procedure, except that learning rate is set as 0.0001 since a larger learning rate makes training PWC-Net unstable.

As shown in Fig. 4, our pre-trained model initially perform on par with PWC-Net on KITTI optical flow training set. After fine-tuning on KITTI-15 stereo images for 75k iterations, although both methods perform similarly on the training data, ours-small gets much lower error on out-of-domain optical flow image pairs. This indicates our model is less overfitted to the training distribution. Qualitative results can be found in the supplementary material.

**Cross-range generalization: small motion $\rightarrow$ large motion** In-the-wild image pairs have unknown maximum displacement, i.e., they may be captured from very different view points and objects can move to anywhere. Therefore, the ability to find correspondences out of training search range is important for real-world applications. To deal with large displacements, one could simply find correspondences on downsampled images. However, this loses high-frequency information. Instead, our proposed separable 4D-convolutional matching module is able to vary search range at test time on demand. To demonstrate this, we train the correspondence model on pixels with small motion (0-32px) on FlyingThings, and test on two displacement ranges (0-32px and 0-64px) on KITTI-15 training set. Ours-32 is our proposed matching module operating on stride 8 features. As a

comparison, we train a PWC-Net baseline using the same annotated data, referred to as PWC-32. We also train a PWC-Net baseline with 0-64px motion to serve as the upper-bound of our method.

As shown in Tab. 3, our method achieves 39.2% lower error than PWC-32 for in-distribution pixels (pixels with 0-32px motion), while achieving 65.4% lower error for out-of-distribution pixels (pixels with 0-64px motion). Moving from in-distribution to out-of-distribution data, the error rate of PWC-32 increases by 231%, while our model increases by 89%, which is on par with a model trained with both in-distribution and out-of distribution data, i.e., PWC-64, demonstrating strong generalization ability to out-of-training-range data.

Table 3: On-demand correspondence matching with extended search range.

| Method | EPE (px) | | ratio |
|---|---|---|---|
| | 0-32px | 0-64px | |
| PWC-32 [38] | 2.85 | 9.44 | 3.31 |
| PWC-64$^{\dagger}$ [38] | 2.72 | 5.50 | 2.02 |
| Ours-32 | **1.73** | **3.27** | **1.89** |

Table 4: Results of single-stage ablation study.

| Method | EPE (px) | GFlops | # Params. |
|---|---|---|---|
| DenseNet [38] | 2.64 | 25.5 | 8.2 |
| Full-4D | 2.30 | 52.5 | 1.83 |
| Sep-4D | 2.31 | 23.4 | 1.78 |
| Ours-UNet | 1.73 | 28.5 | 2.94 |
| UNet→Plain×4 | -0.02 | +20.9 | - |
| - Multi-channel | +0.32 | -0.7 | -0.001 |
| T-soft.→Soft. | +0.10 | -0.5 | - |
| T-soft.→Reg. | +0.58 | -0.4 | +0.001 |
| - OOR | +0.07 | - | - |

## 4.3 Diagnostics

**Single-stage ablation study** To reveal the contribution of each component, we perform a detailed ablation study. For clarity we use a single stage architecture, i.e., without coarse-to-fine warping, on stride-8 features. The models are trained on 0-32px (in both x and y directions) motions on FlyingChairs and evaluated on KITTI-15 training set on pixels with the same motion range. As the baseline model, we implement a DenseNet matching module followed by a refinement module as used in PWC-Net [16], referred to as "DenseNet". For "Full-4D", we replace the DenseNet and refinement module with two residual 4D convolutions blocks (four convolutions in total). As shown in Tab. 4, it reduces error by 12.9% and number of parameters by 77.7%, though with an increased amount of computation. "Sep-4D" separates 4D kernels into WTA kernels and spatial kernels, reducing GFlops by half without significant loss in accuracy. "Ours-UNet" is our final model, which uses multi-channel cost volumes, volumetric U-Net architecture, truncated soft-argmin inference, and out-of-range (OOR) detection. It further reduces the error rate by 23.4%.

We then remove or replace each component from our final model. Replacing the U-Net architecture (ten convolutions) with a plain architecture (eight convolutions) slightly reduces the error but adds a large compute and memory overhead. Replacing the multi-channel cost volume with a standard single-channel cost volume increases the error by 18.5%. Replacing the truncated soft-argmin with a standard soft-argmin increases the error by 6.8%, and direct regression of flow vectors from cost volumes increases the error by almost one-third, demonstrating the benefits of using truncated soft-argmin inference. Interestingly, removing the out-of-range detection module in training also increases error. We posit that it uses knowledge from the cost volume structure to regularize the network and helps the model to generalize better.

**Analysis on cost volume filtering** We also compare different architectural designs of cost volume filtering in terms of FLOPS and numbers of parameters that are used. To filter a multi-channel cost volume of size $(K, U, V, H, W)$, "2D convolution" reshapes the first three dimensions $(k, u, v)$ into a feature vector and filters along the height and width dimension $(x, y)$. Our "4D convolution" and "separable 4D convolution" treat the hypotheses dimension k as feature dimension and filter along the $(u, v, x, y)$ dimension. As shown in Tab. 5, separable 4D convolution uses 3.5X fewer parameters and computations compared to the full 4D convolution. Compared to 2D convolution, separable 4D convolution only uses $\frac{2}{U^2V^2}$ parameters and $\frac{2}{UV}$ computations. Specifically when $U = V = 9$ as in PWC-Net [39], replacing the 2D convolutions with separable 4D convolutions reduces the computation by 40x and number of parameters by 3000x.

Table 5: Comparison between filtering approaches on a (K,U,V,H,W) multi-channel 4D cost volume.

| Method | Kernel | # Param. | ratio | # Mult-Adds | ratio |
|---|---|---|---|---|---|
| 2D conv. | $(KUV, KUV, 3, 3)$ | $9K^2U^2V^2$ | $\frac{U^2V^2}{2}$ | $9HW \times K^2U^2V^2$ | $\frac{UV}{2}$ |
| 4D conv. | $(K, K, 3, 3, 3, 3)$ | $81K^2$ | $4.5$ | $81HW \times K^2UV$ | $4.5$ |
| Sep. 4D conv. | $(2, K, K, 3, 3)$ | $\mathbf{18K^2}$ | $1$ | $\mathbf{18HW \times K^2UV}$ | $1$ |

## 4.4 Stereo matching with vertical disparity

We further show an application of our correspondence network in stereo matching with imperfect rectification. Although most stereo systems assume that cameras are perfectly calibrated and correspondences lie on the same horizontal scan-line. However in reality, it is difficult to perfectly calibrate stereo pairs during large temperature changes and vibrations [12]. Such errors result in ground-truth disparity matches that have a vertical component (e.g., match to a different horizontal scanline). Instead of searching for stereo correspondences along the horizontal scanline, we find matchings in a 2D rectangular area, and project the displacement vector in the horizontal direction.

We fine-tune our model and PWC-Net using stereo data from KITTI, Middlebury, and SceneFlow [9, 29, 30, 35] training set for 70K iterations. For our model, we set $U = 6, V = 1$ for each level. We then evaluate on half-sized Middlebury-14 additional images, where there are thirteen images with perfect rectification and thirteen with imperfect rectification. ELAS [8] is taken from the Robust Vision Challenge official package, and we implemented two-pass SGBM2 [11] using OpenCV (with SAD window size = 3, truncation value for pre-filter = 63, p1 = 216, p2 = 864, uniqueness ratio = 10, speckle window size = 100, speckle range = 32). The results from SGBM2 is also post-processed using weighted least square filter with default parameters.

As shown in Tab.6, going from perfectly rectified stereo images to the imperfectly rectified ones, the error rate of our methods does not increase. While methods without explicit vertical displacement handling, for example, ELAS [8], suffer heavily from such situations. Compared to PWC-Net, our model gets a lower error, possibly due to the effectiveness of volumetric filtering, and is more flexible because of the on-demand selection of search space. A qualitative comparison is shown in Fig. 5. Though ELAS handles stereo images with perfect calibration well, it fails on imperfectly rectified pairs, yielding gross errors on repeated patterns and textureless surfaces as indicated by the circles. Our method is not affected by vertical displacement caused by imperfect rectification, given its pre-defined 2D search space.

Table 6: Results on Middlebury stereo images.

| Method | avgerr (px) | | inc.(%) |
|---|---|---|---|
| | perfect | imperfect | |
| SGBM2 [11] | 14.51 | 15.89 | 9.5 |
| ELAS [8] | 9.89 | 11.79 | 19.2 |
| PWC-Net [38] | 9.41 | 9.92 | 5.4 |
| Ours | **9.03** | **8.79** | **-2.7** |

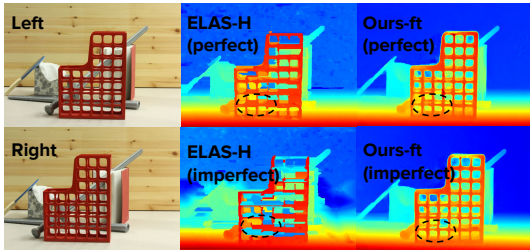

Figure 5: Result on Middlebury-14 image, "Stick2".

## 5 Discussion

We introduce efficient volumetric networks for dense 2D correspondence matching. Compared to prior SOTA, our approach is more accurate, easier to train, generalizes better, and produces multiple candidate matches. To do so, we make use of volumetric encoder-decoder layers, multi-channel cost volumes, and separable volumetric filters. Our formulation is general enough to adapt search windows on-the-fly, allowing us to repurpose optical flow networks for stereo (and vice versa) and implement on-demand expansion of search windows. Due to limited CUDA kernel and hardware support for convolutions and poolings with non-standard shapes, the FLOPS numbers for our current implementation are not directly transferable to running time, which will be explored in the future.

**Acknowledgements:** This work was supported by the CMU Argo AI Center for Autonomous Vehicle Research.

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
