[Supplementary Material]

# Volumetric Correspondence Networks
# for Optical Flow

**Gengshan Yang**[1][*], **Deva Ramanan**[1,2]
[1]Carnegie Mellon University,   [2]Argo AI
{gengshay, deva}@cs.cmu.edu

## 1   Supplementary Material

### 1.1   Detailed training procedure

We re-implement a Pytorch version of the training pipeline of PWC-Net[4], which has three stages: 1) Pre-training on FlyingChairs, 2) fine-tuning on FlyingThings and 3) fine-tuning on the target dataset.

**Pre-training on Chairs**    We use the same hyper-parameters, except that: 1) The network is trained for 140K iterations instead of 1200K iterations, 2) the learning rate is set to be 1e-3 instead of 1e-4, and reduced by half at {70K,130K} iterations, and 3) the weight-decay term is removed.

**Fine-tuning on Things**    We use the same hyper-parameters, except that: 1) The network is fine-tuned for 80K iterations instead of 500K iterations, 2) the learning rate is set as 2.5e-4 and reduced by half at 50K iterations, 3) the weight-decay term is removed, and 4) eight cropped images of size $448 \times 320$ is used in each batch, instead of four cropped images of size $768 \times 384$.

**Fine-tuning on KITTI**    We follow the PWC-Net+ procedure. Besides removing the weight-decay term, we also 1) use a crop of $256 \times 768$ instead of $320 \times 896$, 2) use 4 GPUs with a batch-size of 16 for two cycles, 3) set the initial learning rate as 0.001, and reduce it by half at {30K,40K,50K,60K} iterations for each cycle, and 4) use $L_1$ loss + OOR loss instead of the robust loss function. We also use asymmetric occlusion augmentation as used in HSM-Net [5].

**Fine-tuning on Sintel**    We follow the PWC-Net+ procedure. Besides removing the weight-decay term, we also 1) use a crop of $320 \times 576$ instead of $384 \times 768$ and also add Chairs and Things for training, 2) use 4 GPUs with a batch-size of 16 for two cycles, 3) set the initial learning rate as 0.001, and reduce it by half at {30K,40K,50K,60K} iterations for each cycle, and 4) use $L_1$ loss + OOR loss instead of the robust loss function. We also use asymmetric occlusion augmentation as used in HSM-Net [5].

### 1.2   Details on measuring FLOPS

We use FLOPS (floating point operations per second) to measure the compute needed for a deep network. It is typically calculated by counting the number of Mult-Adds (multiply–accumulate operation) [2, 3, 6]. For example, a dot product with N elements has N Mult-Adds. When computing the FLOPS, we only consider convolutions, batch-norms, activation functions, and pooling layers. The same python package is used to compute the FLOPS for each model [7].

### 1.3   Qualitative results

We further show qualitative results of our flow predictions following the Middlebury color encoding [1].

Figure 1: Qualitative results for stereo → flow transfer. Before fine-tuning, PWC-Net predicts the motion of the left-corner car wrong while our model gets the motion correctly. After fine-tuning, although PWC-Net and ours-small performs similarly on the stereo training data, PWC-Net produces artifect at object boundaries while our model is able to recover the object boundaries clearly.

- Fig. 1 shows the results of "stereo to flow transfer".
- Fig. 2 gives a qualitative comparison between our model and PWC-Net on flow predicted by each scale.
- Fig. 3 shows the K flow hypotheses generated by each level of coarse-to-fine prediction.
- Fig. 4 and Fig. 5 show qualitative comparisons with prior arts on challenging examples from KITTI-15 and Sintel test set.

Figure 2: Comparison of coarse-to-fine prediction between our model and PWC-Net_ROB on Sintel_clean_ambush_2_00. Note that the coarse (stride-64) flow prediction module of PWC-Net is not trainable due to difficulties in optimization. However, our stride-64 matching module is trainable and yields reasonable coarse-prediction. We hypothesize the reason to be that 4D volumetric layers takes advantage of weight-sharing and uses much fewer parameters, which makes gradient propagation easier. Moreover, our model predicts better details as shown in the circle.

Figure 3: Multi-hypotheses predictions for Sintel_clean_ambush_2_00. From top to bottom: hypotheses at stride-32 and hypothese at stride-4.

Figure 4: Results on KITTI-15 test image 48. Color indicates the direction and magnitude of the displacements following Middlebury color wheel, as shown on top-left. While prior methods predict the dark wall (circled) as moving to the right together with the front vehicle, our method correctly predicts it as moving to the left.

Figure 5: Results on Sintel test image bamboo_3_29. Compared with prior arts, our method predicts the motion of the circled region more accurately and captures more details.

## Footnotes

*Code will be available at github.com/gengshay-y/VCN.