[Reviews · NeurIPS 2019]

Reviewer 1



I vote for rejecting this submission from Neurips 2019. For once, it has severe presentation issues. Besides a fairly high number of typos and formatting issues, the explanations in the paper are at times lacking and counter intuitive, especially in the method part 3. The overview over the network architecture in Figure 3 needs to be explained more thoroughly such that it stands for itself. Regarding the method itself, the contribution over prior work is unfortunately not entirely clear to me. The main claim for novelty is an application of a truly 4D cost volume processing in contrast to state-of-the-art methods that "reshape the 4D cost volume as a multichannel 2D array with N = U × V channels" [9,...] (l. 101). However, Flownet [9] which the authors refer to in this context does nothing of the sort, instead they extract patch-wise features for both input images and correlate pairs of patches to get a notion of correspondence. The "offset invariance" (l. 104) the authors claim to have in comparison to prior work is therefore not novel but rather predominant in state-of-the-art methods in this field. Furthermore, in order to reduce the computational load the authors are separating 4D convolutions into two 2D operations (l. 123) acting independently on the two image domains. It seems to me that this allows for only very limited correlated interactions that are able to take advantage of the authors claims of using "true 4D operations". Furthermore, given that the set of core operations (2D convolutions) to fuse information is similar, it seems to me that overall this approach has a strong theoretical connection with the strategy in Flownet or other state-of-the-art methods. This aspect needs to be explored and clarified by the authors, otherwise the contribution over prior work is dubious to me. I find the evaluations in the paper lacking and not in coherence with the authors claims of "dramatically improving accuracy over the state-of-the-art". Qualitative comparisons are missing completely in the paper, there should be at least one depiction of how the method improves upon prior work on some challenging example. While the authors provide a couple of examples in the supplementary material, those comparisons are all done wrt one other method only (PWCnet).

Reviewer 2



Originality: This paper consists of several engineering "tricks" that enable 4D cost volume filtering. While these modifications re relatively straightforward, they are well motivated, principled and of great use to practitioners and deployment of optical flow models. Quality: The technical content of the paper appears to be correct. There are multiple aspects I like about this paper: + Proposed modifications seem to speedup the training process (~10x) and improve its stability (wider range of learning rates). + Proposed model significantly reduces number of required flops (< 50% of PWCNet, ~25% of LiteFlowNet), and memory while achieves state of the art accuracy (among compact models). + One of the motivations of this paper is to reduce memorization and improve generalization - this is nicely demonstrated on two tasks (stereo -> flow and small -> large motion). Perhaps, it might be interesting to see whether it also helps domain adaptation (however this is certainly beyond the scope of this paper). + Experiments in this paper are very clearly described and contain great discussion. + As this paper would be of great use to practitioners, its also nice to see the authors will release the code. Clarity: The paper is very-well written, contains nice figures that illustrates well the key concepts and authors will release the code. Perhaps, I'd just modify lines 22-30 which somewhat repeat the abstract and certain design choices could be better motivated/explained/justified (e.g. why cosine similarity is used). It is well organized, experiments section contain great discussion and I've enjoyed reading it! Significance: Memory and compute requirements of modern optical flow estimation methods represent a real bottle-neck - this paper might make deployment of optical flow much more easier. This paper proposes several relatively straightforward modifications of standard dense optical flow matching models, however, I believe these would be of great use to practitioners!

Reviewer 3



This submission presents a number of modifications to simplify volumetric layers in networks for optical flow estimation. These not only improve the memory and execution time requirements, but also accuracy when compared to previous work. It is also demonstrated that the new networks are able to more effectively generalize to and be repurposed for other correspondence tasks. Originality: This is a new method for the visual correspondence problem, and addresses quite a few shortcomings or limitations of existing methods. Related work is adequately cited and used to contextualise the proposed work. Quality: The authors takes care to properly explain and motivate design choices, and back up claims with a sufficient amount of experimental evidence. They are also successful in evaluating the strengths of the method, but could have perhaps put more effort into analysing its weaknesses. Clarity: The submission is written and organized well. I only found a few small typos, as listed under "improvements" below. Significance: This work does seem to be significantly new, with important results that can advance state-of-the-art research and practice of optical flow estimation. Tests show quite conclusive improvements over previous work.

[Author Response · NeurIPS 2019]

We thank reviewers for the constructive feedback. Reviewers think our paper is "very-well written" (R2), "significantly new" (R3), and "of great use to practitioners" (R2, R3). Some pressing concerns are addressed below.

**Contribution & separable filters (R1,R3)** R1 states that our contribution over prior work, for example, FlowNet [9], is not clear. Specifically, we claim that prior deep flow networks (including [9]) re-organize the 4D cost volume into a 3D array that is processed with multi-channel 2D convolutions, but R1 states that [9] "does nothing of the sort". We humbly disagree. The following is a direct quote from [9]:

> In theory, the result produced by the correlation is four-dimensional: for every combination of two 2D positions we obtain a correlation value, i.e. the scalar product of the two vectors which contain the values of the cropped patches respectively. In practice we organize the relative displacements in channels. This means we obtain an output of size $(w \times h \times D^2)$.

The re-organized costs are then convolved along the $(w, h)$ dimensions, leaving the filtering across displacement dimensions to be fully-connected (FC). Instead, we leave the $(w \times h \times D \times D)$ cost volume as-is and directly apply 4D convolutions. R1 asks how separable 4D filters differ from 2D multi-channel FC filters. Separable 4D filters have far fewer parameters (8.2M→1.78M weights, Tab. 5), increasing efficiency and generalizability. Specifically, because (separable) 4D filters are fully-convolutional, they can be trained models with one displacement and applied at differing ones (e.g., repurposing stereo networks for flow and vice-versa). This is difficult for FC layers.

Figure 1: Illustration of volumetric processing at one pyramid level. 1) Cost volume construction: We warp features of the target image using the upsampled coarse flow and compute a multi-channel cost volume. 2) Volume processing: The multi-channel cost volume is filtered with separable 4D convolutions, which is integrated into a volumetric U-Net architecture. We predict multiple flow hypotheses using truncated soft-argmin. 3) Soft selection: The flow hypotheses are linearly combined considering their uncertainties and the appearance feature.

**Revision (R1,R2,R3)** The revised architecture plot is shown in Fig. 1. We will improve the readability of the method section, fix typos/format issues, and incorporate other feedback (e.g., analysis of the weakness) in the camera-ready.

**Qualitative comparisons (R1)** For other baselines besides PWC-Net+, we show qualitative results of a challenging example from KITTI-15 test set in Fig. 2. We will add more qualitative results to the main paper and project website.

Figure 2: Results on KITTI-15 test image 48. Color indicates the direction and magnitude of the displacements following Middlebury color wheel, as shown in the top-left image. While prior methods predict the dark wall (circled) as moving to the right together with the front vehicle, our method correctly predicts it as moving to the left.

**Variations of results (R3)** We run five trials of the pre-training stage of "Ours-small" model and compute the mean/std of EPE on KITTI, which becomes $9.43\pm0.18$ (used to be 9.31 in Tab. 3). More results will be added to the camera-ready.

[Meta-Review · NeurIPS 2019]

• This paper presents a method for dense optical flow estimation Proposing a 4D method capable to reduce the significant amount of memory. The proposal is very efficient in term of speed and it will be very useful in computer vision. • The three reviewers were not concordant in term of rate ( 1 reject and two accept rates) • After the rebuttal the two positive reviewers were satisfied by the rebuttal keeping positive the rate. The first reviewer still was not convinced about the novelty and the clearness of the method. After a discussion the area chair suggest an acceptance.